# Cell-Based Glioma Models for Anticancer Drug Screening: From Conventional Adherent Cell Cultures to Tumor-Specific Three-Dimensional Constructs

**DOI:** 10.3390/cells13242085

**Published:** 2024-12-17

**Authors:** Daria Lanskikh, Olga Kuziakova, Ivan Baklanov, Alina Penkova, Veronica Doroshenko, Ivan Buriak, Valeriia Zhmenia, Vadim Kumeiko

**Affiliations:** School of Medicine and Life Sciences, Far Eastern Federal University, 690922 Vladivostok, Russia; rikhiks@gmail.com (D.L.); olga.yu.kuziakova@gmail.com (O.K.); che1p4uk@gmail.com (I.B.); palina2609@gmail.com (A.P.); doroshenko.vo@dvfu.ru (V.D.); cutekasatik@gmail.com (I.B.); valeri3066@gmail.com (V.Z.)

**Keywords:** glioma, three-dimensional model, brain tumors, cancer, genetically modified model, cell-based screening, patient-derived culture, organoid, spheroid, early passage cells

## Abstract

Gliomas are a group of primary brain tumors characterized by their aggressive nature and resistance to treatment. Infiltration of surrounding normal tissues limits surgical approaches, wide inter- and intratumor heterogeneity hinders the development of universal therapeutics, and the presence of the blood–brain barrier reduces the efficiency of their delivery. As a result, patients diagnosed with gliomas often face a poor prognosis and low survival rates. The spectrum of anti-glioma drugs used in clinical practice is quite narrow. Alkylating agents are often used as first-line therapy, but their effectiveness varies depending on the molecular subtypes of gliomas. This highlights the need for new, more effective therapeutic approaches. Standard drug-screening methods involve the use of two-dimensional cell cultures. However, these models cannot fully replicate the conditions present in real tumors, making it difficult to extrapolate the results to humans. We describe the advantages and disadvantages of existing glioma cell-based models designed to improve the situation and build future prospects to make drug discovery comprehensive and more effective for each patient according to personalized therapy paradigms.

## 1. Introduction

Gliomas remain one of the most aggressive and therapeutically challenging brain tumors. Despite advances in understanding their molecular underpinnings, treatment options for glioma patients remain limited, with poor prognosis and low survival rates, especially for high-grade tumors. The complexity of gliomas, characterized by their genetic heterogeneity, infiltrative growth, and resistance to conventional therapies, complicates both diagnosis and treatment [1,2]. Preclinical glioma research has explored various approaches, each offering valuable insights into the biology of gliomas and the development of effective therapies. However, each of these models has limitations, and a more comprehensive approach is needed to capture the full complexity of glioma biology and improve drug discovery.

Genetically modified glioma models, including transgenic mouse models, have played a pivotal role in replicating the key genetic alterations found in human gliomas, such as mutations in EGFR, p53, and *IDH1*. These models enable the study of specific genetic mutations and their contribution to tumor initiation and progression [3,4]. While they offer valuable insights into glioma biology, genetically modified models often fail to reflect the full diversity of gliomas, particularly their interactions with the surrounding microenvironment and the heterogeneity present within individual tumors. Thus, while these models are essential for understanding the molecular mechanisms driving gliomas, they are not sufficient on their own to represent all aspects of glioma biology.

Another major advancement in glioma modeling is the use of three-dimensional (3D) cell cultures, such as organoids and spheroids, which provide a more physiologically relevant platform compared to traditional two-dimensional (2D) cultures. Three-dimensional models allow for the study of key tumor characteristics, including cellular heterogeneity, tissue architecture, and interactions with the extracellular matrix, all of which are critical to tumor progression and therapy resistance [5,6]. These models have proven especially useful in testing drug efficacy, particularly for targeted therapies and immunotherapies, by more closely mimicking the in vivo tumor environment. However, 3D models also face challenges, including difficulties in scaling and maintaining long-term culture conditions, which can limit their widespread application in preclinical drug testing.

Here, we describe conventional adherent glioma cell lines, patient-derived models, genetically and epigenetically modified cell models, and 3D cell-based constructs. This review aims to explore the strengths and weaknesses of current glioma models, discuss how these models contribute to the development of personalized therapeutic approaches, and highlight emerging technologies that may offer more comprehensive insights into glioma treatment. In order to reflect on integrating various modeling strategies, researchers can enhance the predictability of drug responses and ultimately improve the clinical management of glioma patients.

## 2. Molecular Features of Gliomas

Gliomas are the most common malignant tumors in the central nervous system (CNS) originating from glial cells. The current fifth edition of the WHO Classification of Tumors of the Central Nervous System was published in 2021 and is focused on molecular markers in diagnostics, prognosis, and treatment of brain tumors [7].

There are several groups of glioma included in the latest classification of CNS tumors: (1) adult-type diffuse gliomas, (2) pediatric-type diffuse low-grade gliomas, (3) pediatric-type diffuse high-grade gliomas, (4) circumscribed astrocytic gliomas.

Glioma types differ significantly in their molecular signatures, morphological characteristics, and patient prognosis. These differences must be considered when developing models to study carcinogenesis mechanisms and assess drug efficacy. Treatment effectiveness can vary greatly depending on the molecular features of the tumor and the disease stage, highlighting the need for specific models for each type of glioma, which can be useful for diagnosis, predicting treatment response, and developing personalized therapeutic strategies.

Adult-type diffuse gliomas include astrocytoma, isocitrate dehydrogenase mutant (*IDH*-mutant); oligodendroglioma, *IDH*-mutant and *1p/19q*-codeleted; and glioblastoma with *IDH*-wildtype. The first-line diagnostic marker is *IDH1/2* status. DNA sequence variants in the key sites of *IDH1/2* are favorable prognostic markers of low-grade gliomas (LGG) and lead to neomorphic enzymatic activity, where normal alpha-ketoglutarate (a-KG) is converted to oncometabolite D-2-hydroxyglutarate (2-HG) with the consumption of one molecule of NADH. Elevated levels of 2-HG induce epigenetic changes, including DNA and histone hypermethylation, dysregulation of gene expression, and activation of HIF signaling, and accumulation of NAD+ leads to oxidative stress [8,9]. Other genes that might be altered in *IDH*-mutant astrocytoma are *ATRX*, *TP53*, and *CDKN2A/B* [7].

It is important to emphasize that all the previously mentioned molecular markers of different types of gliomas, as well as the characteristics of the molecular functioning of glioma cells and their interaction with the microenvironment described below, represent potential therapeutic targets. The development of glioma cell models must consider the therapeutic agent being studied to ensure an accurate representation of drug effects and key gliomagenesis factors. Well-designed models should comprehensively reflect the complexity of gliomas, enabling precise reproduction of pathological processes and supporting the development of effective treatment strategies.

Oligodendroglioma is characterized mainly by the presence of alterations in the *IDH1/2* sequence and chromosomal codeletion of *1p/19q*. A combination of both markers leads to significantly better survival in patients. A possible reason for the favorable prognosis could be the suppression of a tumor-immune microenvironment in gliomas with *1p/19q* loss [10]. Other DNA sequence variants important for oligodendroglioma diagnostics are in the telomerase reverse transcriptase (*TERT*) promoter, *CIC*, *FUBP1*, and *NOTCH1* [7].

The absence of the altered DNA sequence variants of *IDH1/2* in gliomas is a sign of high grade (HGG). Glioblastoma is a rapidly proliferative high-grade glioma with a poor prognosis. The molecular profile of this tumor includes pathogenic polymorphisms in the epidermal growth factor receptor (*EGFR*), *TERT* promoter, and 7 and 10 chromosomal alterations. Different mutations in the *EGFR* extracellular domain make the receptor constitutively active, which means ligand-independent phosphorylation of downstream targets from MAPK (mitogen-activated protein kinase), JAK (Janus kinase), STAT (signal transducer and activator of transcription), PLCγ (phospholipase Cγ), PKC (protein kinase C), PI3K (phosphoinositol-3 kinase), and Akt (or PKB, protein kinase B) pathways. Despite the variety of EGFR inhibitors, glioblastoma-targeting anti-EGFR therapy remains challenging due to the low mutational burden, specific alterations in glioblastoma, and the blood–brain barrier [11].

Circumscribed astrocytic gliomas include less aggressive gliomas—pilocytic astrocytoma, pleomorphic xanthoastrocytoma, subependymal giant cell astrocytoma, chordoid glioma, and *MN1*-altered astroblastoma. These gliomas frequently have a favorable outcome after surgical resection [12]. However, there is also high-grade astrocytoma with piloid features. The molecular markers of circumscribed astrocytic gliomas are *NF1*, *BRAF*, *MN1*, *CDKN2A/B*, *ATRX*, *TSC1/2*, and *PRKCA* [7,13].

The pathophysiological process in glial cells leading to malignancy includes many steps in which normal cells undergo genetic alterations, changes in gene expression, loss of cell cycle control, and over-expression of growth factor receptors, which results in angiogenesis, invasion, migration, genetic instability, and disorder of apoptosis (Figure 1) [14]. There are several key elements that could be used as therapeutic targets: altered molecular pathways, epigenetic dysregulation, chromosomal instability (CIN), microenvironment disorganization, and immune resistance.

The most frequently altered molecular pathways in gliomas are the p53 pathway, PI3K-Akt, the Rb pathway, and Ras/Raf/MAPK [15]. These pathways are involved in proliferation, cell growth, migration, apoptosis, and cell survival. The Wnt/β-catenin pathway is considered to be responsible for the invasion and migration of gliomas [14]. In addition, there is a JNK signal pathway with a dual role in cell death and cell survival in CNS; however, in gliomas, JNK activation is associated with cancer stem cell-like properties, cancer-initiating potential, and glial proliferation [16]. On the other hand, Li et al. identified the activation of the JNK signal pathway as a key target of MOMIPP (cytotoxic indolyl-pyridinyl-propenones compound)-induced methuosis (non-apoptotic cell death) in glioblastoma [17].

The invasion phenotype correlates with the expression of growth factors and/or their receptors (VEGFA (vascular endothelial growth factor A), EGFR, PDGFR (platelet-derived growth factor receptor), cell-adhesion receptors (Eph/Ephrin family, CD44), and proteases (serine proteases, cathepsins, MMPs, and the ADAMTS family of metalloproteases)) [18]. Focal adhesion kinase (FAK) takes part in the activation of various signal pathways emanating from integrins and growth factor receptors. FAK could be considered as a prospective target for therapy, since its activation stimulates proliferation, migration, and invasion of tumor cells. Alza et al. successfully arrested the glioblastoma cell cycle through the inhibition of FAK [19].

A characteristic feature of solid tumors is the occurrence of hypoxia and, as a result, inhibition of the activity of prolyl hydroxylase enzymes (PHDs), which leads to activation of the HIF-1 transcription factor. The consequences of activation are associated with a poor prognosis of the disease, since HIF-1 induces the expression of many proteins, including the proangiogenic factors VEGF, PDGF, EPO, and Ang-1/2, stimulating angiogenesis [20,21]. Metabolic reprogramming of tumor cells also occurs due to the transcription of the glucose transporter GLUT1/3, lactate dehydrogenase LDH-A, and monocarboxylic transporter 4 (MCT4), and modulation of lipid metabolism [22]. In general, HIF-1 stabilization in glioma cells leads to a more aggressive tumor phenotype, increased vascularization, and greater invasiveness due to the initiation of the epithelial–mesenchymal transition.

Epigenetic changes in gliomas include DNA methylation, histone modification, and chromatin remodeling [23]. *IDH*-mutant gliomas produce 2-HG oncometabolite that leads to DNA methylation patterns, named glioma CpG island methylator phenotype (GCIMP) [24]. Additionally, deregulation of miRNA in tumor cells results in uncontrolled expression of target genes involved in cell proliferation, resistance to apoptosis, autophagy, invasion and metastasis, angiogenesis, and drug resistance. As there are both oncogenic and anti-oncogenic miRNAs, they could promote glioma-cell proliferation (miR-21) or be used as therapeutic molecules (miR-128, miR-34a) [25].

The glioma microenvironment is formed by cellular and acellular components, which contribute equally to gliomagenesis, tumor progression, invasion, and therapy resistance. Cellular components include glial cells, neurons, and inflammatory cells (microglia, monocytes, and macrophages). Astrocytes are involved in ion and neurotransmitter homeostasis. Peritumoral astrocytes have decreased this function, which results in high glutamate concentration in the glioma microenvironment that promotes proliferation [26]. Furthermore, 2-HG may mimic glutamate activity [27]. Neurons promote the release of the activity-dependent mitogens neuroligin-3 (NLGN3) and brain-derived neurotrophic factor (BDNF). Direct electrochemical communication promotes tumor invasion with AMPARs (α-amino-3-hydroxy-5-methylisoxazole-4-propionic acid receptors, important in fast neurotransmission) on glioma cells. Non-neoplastic astrocytes facilitate tumor growth, realizing anti-inflammatory cytokines such as transforming growth factor beta (TGF-β), interleukin 10 (IL-10), and granulocyte colony-stimulating factor (G-CSF), and promote expression of HIF-1a [28]. The immune microenvironment predominately consists of tumor-associated macrophages/microglia (TAMs), and also of macrophages, microglia, myeloid-derived suppressor cells, lymphocytes (CD8+ cytotoxic T cells, CD4+ regulatory T cells, and B cells), natural killer (NK) cells, and neutrophils.

Single-cell studies revealed new potential molecular signatures influencing the glioma microenvironment. Elevated CD147 expression in grade 3 gliomas and M2 macrophages, key components of the glioma microenvironment, plays a crucial role in promoting tumor proliferation and inhibiting cancer cell apoptosis [29]. TIMP1 inhibition was shown to suppress LGG cell proliferation, migration, invasion, and the polarization of tumor-associated macrophages [30]. Additionally, ELF4 was found to drive the reprogramming of tumor-associated monocytes and macrophages (TAMMs) [31]. FAM109B is another novel prognostic biomarker for low-grade gliomas, which exhibits specific overexpression in TAMs and may be a potential therapeutic target for LGG patients [32].

As for the acellular component of the tumor microenvironment, fibrillar proteins, such as collagen, fibronectin, laminins, and periostin, predominantly make the ECM (extracellular matrix) relatively stiff, while normal brain ECM is relatively soft because of glycosaminoglycan (GAG)-containing molecules [28]. Other important components include proteoglycans, glycoproteins, and degradative enzymes.

There are two types of proteoglycans upregulated in the glioblastoma ECM relative to normal tissue—heparan sulfate proteoglycans (HSPGs) and chondroitin sulfate proteoglycans (CSPGs) [33]. Several studies have shown the correlation between matrix sulfate rate and glioma invasiveness [34,35,36]. Another important linear non-sulfated glycosaminoglycan component is hyaluronic acid (HA). Glioma cells on the migration front express CD44 as a main surface receptor to HA and other hyaluronan-binding proteins—RHAMM and BEHA. Interaction between HA and tumor cells facilitates cell migration and promotes tumor invasion. Furthermore, CD44 binds the matrix metalloproteinase (MMP) MMP9, and this interaction causes matrix degradation and provides migration routes [37].

Fibrillar proteins (collagen, fibronectin, laminins, and periostin) are not abundant in normal brain matrix but increase in the glioma ECM, predominantly around vessels providing tumor cells adhesion and migration [33].

The concentration of tenascin in the glioma ECM is associated with increased adhesion to the ECM and facilitates cell migration but not proliferation [38].

Integrins are members of transmembrane adhesion receptors; they provide bidirectional crosstalk between cells and the surrounding components of the ECM. There are different integrins for various ligands—collagen, laminin, and tenascin [39]. The expression level of integrins correlates with the malignancy of glioma cells [40]. The variety of integrins and their role in gliomagenesis make them good therapeutic targets against gliomas [39].

To sum up, gliomagenesis is a long and complex process involving multiple steps. Various therapeutic strategies should be considered and modeled to (1) affect signaling and metabolic pathways, (2) remodel the ECM, and (3) modulate intercellular interactions [41]. All promising treatment strategies need to be tested on various in vitro models to fully assess their therapeutic significance and other relevant properties. Here, we describe conventional adherent glioma cell lines, patient-derived models, genetically and epigenetically modified cell models, and 3D cell-based constructs. We also suggest testing marine-derived drugs on these models to evaluate their potential efficacy in targeting key pathways.

## 3. Glioma Cell Lines in Cell Culture Collections: Variety, Strengths, and Weaknesses

Cell lines are the simplest models for studying the mechanisms of gliomagenesis and drug screening. Cell lines are frequently employed in a range of studies due to their capacity for indefinite cultivation. However, it is important to note that they may potentially lose the characteristics of the original tissue [42]. The most utilized glioma cell lines are U87MG, U251MG (U373MG), T98G, and U118MG (U138MG) [43,44].

U87MG, together with U251MG (U373MG) and U118MG (U138MG), were obtained more than 50 years ago at Uppsala University in Sweden. U87MG is a cell line derived from human glioblastoma and has epithelial cell-like morphology and a hypodiploid karyotype [43]. After establishing U87MG, they were found to highly express cyclin E, c-Myc, CDK4, MDM2, and wild-type p53 [45,46]. A series of genetic profiling experiments were then performed on U87MG, the mutational profile of which revealed multiple homozygous mutations, indels, microdeletions, and interchromosomal translocations (reciprocal translocations between a small fragment of chr1 with chr2 and chr16 at the lower ends) [47]. However, despite the many mutations, the U87MG cell line exhibits chromosomal stability and proliferative activity compared to other lines [48].

U251MG is a glioma cell line and has epithelial morphology [49]. U251MG cells carry mutations in the *TP53* gene and loss of heterozygosity in chromosome 10, which includes the *PTEN* gene [50,51]. U251MG also highly expresses vimentin and GFAP [52]. This cell line is frequently used in xenotransplantation experiments due to its high tumorigenic capacity [53,54].

U138MG and U118MG are glioma cell lines, which are cytogenetically identical and have a common number of tandem repeats [55]. Both cell lines were derived from GB. U138MG differs from U87MG in its low growth rate and hyperploid chromosome set [48]. U118MG has a polyploid karyotype [43].

A172 and T98G are human glioblastoma cell lines. These lines are polyploid [43,44,56]. T98G has a fibroblast-like morphology. The T98G cell line highly expresses the *ACTA2* gene, which encodes alpha-smooth muscle actin (α-SMA), which is involved in cell motility and structure [57]. A172 cells highly express the mesenchymal markers CD90 and CD105, fibroblast activation protein, and tenascin. These cell lines are used to test therapeutic agents [58,59].

The use of cell lines in research is considered the gold standard, but what are their advantages and disadvantages? One of the main disadvantages is the difference in genotype and phenotype compared to the original tissues [60]. It has been shown that with increasing cell passage, the genetic, epigenetic, and morphological profiles change [56,61,62,63]. In addition, there is a lack of systematic screening of cell lines and their comparison with the patients’ cells characteristics [64,65,66].

Media with different compositions can be used for culturing glioma cell lines. The neurobasal medium is a special one for brain cells based on DMEM with reduced glutamine content and low osmolarity. It is optimized for serum-free cultivation, with the addition of growth supplements such as B27 and N2 [67,68]. Unfortunately, the use of serum-free cultures leads to a decrease in proliferation compared to cultivation in FBS-supplemented medium [69].

Therefore, DMEM or DMEM/Ham’s F12, supplemented with FBS and a high glucose concentration, is an optimal medium for glioma cell line cultivation, as FBS provides more nutrients essential for cell growth and division [70]. Nevertheless, this approach has limitations, largely due to the influence of FBS on the genotype and phenotype of cells. FBS may contain extracellular vesicles, RNA, protein aggregates, and calcium oxalates. Those extracellular vesicles may carry nucleic acids, proteins, and lipids, which are transferred into cells and can change their genotype, phenotype, and functional properties and affect intercellular communication [71,72,73,74].

Cell lines are easily subjected to genetic modifications, and the efficiency of obtaining modified lines is higher than when using cells with a low passage [75,76]. And finally, they are simple in cultivation, as written previously, which does not entail high costs [77].

## 4. Patient-Derived Low-Passage Cell-Based Models

Although various glioma cell models are based on conventional long-term cultivated cell lines, the most accurate representations of the real conditions are the models created from patient-derived low-passage cell cultures. There are numerous methods for obtaining cell cultures from various organs and tumor types [78,79,80,81]. For glioma samples, most methods involve the enzymatic digestion of the tumor using collagenases I, II, IV, and trypsin and/or mincing with a scalpel, followed by passing the homogenate through a cell strainer to obtain a cell suspension. Then, obtained cells are cultivated in Neurobasal or DMEM/Ham’s F12 culture medium in the presence of fetal bovine serum (FBS) or with the addition of various growth factors and supplements such as EGF, bFGF, HBEGF, PDGF, B27, and N2 [69,74,82,83].

The most valuable application of patient-derived cell cultures is treatment modeling, which encompasses the evaluation of responses to various drugs and radiotherapy protocols [84]. The cells obtained from patient tumors include various cell types, which could vary in their sensitivity to different therapeutic approaches. The main advantage of the patient-derived glioma cell models is the opportunity to test therapies on tumor-derived cells, which, as demonstrated in single-cell studies, have high heterogeneity, rather than on permanent cell lines [85,86]. Besides advancing cancer research, therapeutic testing results in clinical practice can facilitate the development of personalized strategies and could be used for the therapy selection for each patient, thus driving progress in precise medicine (Figure 2) [87,88].

Low-passage cancer cell lines are typically preferred because they more closely reflect the characteristics of the original tumor, whereas prolonged culturing can lead to changes in the genetic and molecular profiles [88]. Moreover, it is described that cultivation of patient-derived glioma cells in serum-free, EGF/FGF-2-supplemented Neurobasal medium retains the similarity of cells to the parental tumors more effectively than those cultured in serum-containing DMEM [74]. This makes low-passage patient-derived cell cultures very valuable for screening for sensitivity to diverse therapies.

Johansson et al. created a Patient-Derived Cell Atlas that provides information about drug responses in 100 patient-derived human glioma cell cultures by detecting associations between hallmark pathways and multiple drug classes, thus offering a starting point for research into drug repurposing and precision therapy. Usually, most researchers test therapeutics in combination with standard treatment (temozolomide and radiotherapy) to find the approach that enhances the treatment effect and overcomes resistance [89]. Riess et al. provided evidence for the antitumoral effects of mono- and dual treatment with cyclin-dependent kinase inhibitors on low-passage GB models and identified mechanisms of response. Notably, all patient-derived GB cell lines tested in this study were sensitive to abemaciclib and dinaciclib, while the overall response to palbociclib was weaker and additionally cell line-specific [90].

But unfortunately, for the development of the newest therapeutics, permanent glioma cell lines still remain more accessible, as they grow much faster than low-passage glioma cell cultures and are easier to obtain [91].

Despite numerous studies that show different approaches to obtain and use patient-derived glioma cells for treatment modeling, most of them describe only glioblastoma features, lacking the exhibition of other types of gliomas such as *IDH*-mutant astrocytoma and oligodendroglioma [92]. Since low-grade gliomas inevitably progress to secondary glioblastomas, it is crucial to evaluate the role of *IDH*-targeted therapy to improve patient outcomes [93]. *IDH*-mutant gliomas are one of the very few tumors that allow for targeted therapies. A key requirement for evaluating drug efficacy is comparing results to an appropriate control group. However, existing outcome data often combine *IDH*-mutant and wild-type tumors. To design effective clinical trials for *IDH* mutant gliomas, it is crucial to match anamnesis and outcomes specifically for each patient [94]. Small-molecule inhibitors of the mutant *IDH* enzyme and other drugs targeting altered pathways are progressing from preclinical to clinical trials. Miller reviewed existing approaches targeting *IDH*-mutant gliomas only. The main groups include *IDH* inhibitors, demethylating agents, PARP inhibitors, CDK4/6 inhibitors, glutaminase inhibitor, dihydroorotate dehydrogenase (DHODH) inhibitor, immune checkpoint inhibitor, and even vaccines [95]. Large-scale drug screening in patient-derived *IDH*-mutant gliomas identified seven FDA-approved drugs that are active against *IDH*-mutant gliomas (bortezomib, carfilzomib, daunorubicin, doxorubicin, epirubicin, omacetaxine, and plicamycin) [96].

One of the reasons for the lack of models of *IDH*-mutant patient-derived glioma cell cultures is that since such cell cultures have low-grade characteristics, they proliferate reluctantly and are non-effective. Luchman et al. obtained a tumor resection from a patient with grade 3 *IDH*-mutant astrocytoma and established a permanent cell line that preserved the *IDH1* mutation and demonstrated self-renewal and multipotent capabilities [97]. BT142 is the first brain tumor cell line with an endogenous *IDH1* mutation and detectable 2-HG production both in vitro and in vivo. It is widely used in research to advance the understanding of various conditions and develop effective treatments [98,99,100,101,102]. Rohle et al. obtained another significant patient-derived *IDH*-mutant cell line, TS603. It was derived from a patient with anaplastic oligodendroglioma (WHO grade 3) and carries a *1p/19q* codeletion [103]. Kelly et al. described obtaining established oligodendroglioma cell lines containing t(1;19)(q10;p10) that maintain the genetic signature of the parent oligodendrogliomas, express neural stem cell markers, and grow as multipotent oligodendroglioma spheres [104].

The main disadvantage of patient-derived low-passage cell cultures is that obtaining them presents significant challenges, starting with the need for ethical approval to work with human postoperative material and extending to difficulties in establishing standardized protocols for cell isolation, as well as maintaining their proliferation beyond 1–2 passages [105]. Another disadvantage concerns patient-derived 2D adherent cell cultures. Despite harboring the original genetic characteristics of the tumor, 2D models fail to replicate the complex architecture and tumor microenvironment of the human brain, limiting their ability to model critical physiological barriers encountered by drugs, such as tumor hypoxia and impaired diffusion [106,107]. Primary cell culture models fail to replicate the unique composition of the brain, which exhibits high heterogeneity that could be lost in the early stages of culture due to the selective growth of more adaptable cell populations or even be overgrown by stromal components, which may outcompete cancer cells and compromise the representativeness of the culture model [96]. This significantly reduces their applicability in preclinical drug development and testing.

Along with that, glioma cells, like any other patient-derived cells, may require a complex culture medium, often containing growth factors and specific supplements, to support growth and maintain their phenotype.

Recent advances have led to the development of 3D constructs known as organoids, which closely mimic human tissues and organs. These organoids address the limitations of early glioma models and are now the most promising platform for glioma research [108,109,110].

Jeising et al. established a patient-derived primary GB sphere culture and showed that the EC50 for verteporfin in photodynamic therapy in glioblastoma for that cell culture was much higher compared to the established cell lines [111]. Another study showed that a patient-derived glioblastoma model with intra-tumoral heterogeneity and resistance to conventional therapy cultivated as 2D culture was sensitive to vismodegib, disulfiram, parthenolide, omipalisib, and costunolide. However, in 3D culture, only the TERT inhibitor costunolide was effective against it [112].

For studying low-passage *IDH*-mutant gliomas, patient-derived xenograft (PDX) models are one of the most effective approaches for investigating their properties. This approach involves forming spheroids or organoids from patient-derived tumor cells, which are subsequently transplanted into the brains of immunocompromised animals [93,113,114,115,116,117,118,119].

Jacob et al. presented a rapid method for creating and biobanking patient-derived glioblastoma organoids that accurately replicate tumor features and mutational profiles by microdissection of resected tumors without mechanical or enzymatic dissociation of the tissue into cell suspension [79]. Tumor-derived organoids can be used for testing the efficacy of anti-tumor drugs, existing, new, and off-label, as well as for evaluating personalized CAR T cell therapy by modeling tumor-immune interactions. Additionally, these organoids provide a platform for studying virotherapy and radiotherapy responses in a physiologically relevant 3D environment, allowing for more accurate predictions of therapeutic outcomes in individual patients [120,121,122,123].

## 5. Genetically Modified Glioma Cells for Precise Targeted Drug Screening

There is a class of brain cancer models that have been generated by genetic modifications (Figure 3). The first attempt to obtain a cell model was the introduction of SV40 T-antigen into the 5′ flanking sequence of the murine glial fibrillary acidic protein (GFAP) promoter and the successful malignanization of mouse astrocytes [124]. Later, attempts were made to introduce oncogenes such as EGFR and CDK4 by somatic cell gene transfer using viral vectors based on replication-competent avian leukosis virus splice acceptor (RCAS) and their TVA receptors to generate glioma models [125]. Various glioblastoma models have been generated by introducing oncogenes such as *Src*, *K-ras*, *H-ras*, *PDGFB*, and Egfr vIII [126,127]. However, it remains an open question whether the findings from these studies are relevant to human disease.

To eliminate the biological differences characteristic of murine models, human cell-based models are required. Rich et al. [128], followed by Sonoda et al. [129], engineered human astrocytes with a combination of TERT and HRAS expression and p53 pathway inhibition via viral transduction and successfully established high-grade glioma models. These models facilitated investigations into the mechanisms of gliomagenesis in a human cellular context. But several characteristics of clinical glioblastoma have yet to be subjected to rigorous investigation in these models, including inter- and intratumor heterogeneity.

Additional cases include various monolayer cell lines derived from human gliomas [60]. Nevertheless, despite the extensive data amassed from these cell lines, it remains challenging to ascertain their clinical relevance. This is especially the case in light of the emergence of mutations since the isolation of the original cells, as well as the persistent inadequacy in recapitulating the heterogeneity observed in clinical glioblastoma [130].

### 5.1. Models for Studying Driver Mutation Influence on Glioma Pathogenesis

The main driver mutations, the effect of which has been studied in genetically modified models, are oncogenic mutations in *EGFR*, *RAS,* and in the suppressor genes *PTEN*, *TP53*, *CDKN2A*, and *CDKN2B* [131]. These genetic changes affect four main molecular cascades: Ras-Raf-MEK-ERK, PI3K/AKT/mTOR, the p53 pathway, and the Rb pathway [132].

For example, early studies were devoted to the creation of lines constitutively expressing the mutant EGFR variant (ΔEGFR/EGFR vIII) based on glial cells using the RCAS vector system [125]. This model was used to study the role of mutation in *EGFR* in the induction of glioma formation. The human glioma cell line U373MG, which expresses ΔEGFR or its kinase-deficient mutant with the EGFP reporter, shows the dynamics of glioma growth and its dependence on the corresponding mutations, as well as an increased expression of the *KLHDC8A* gene associated with the suppression of EGFR activity, which is presumably responsible for tumor recurrence [133]. Recent developments in this field have led to the creation of a conceptually similar model on LNT-229 and U87MG cells with different *EGFR* statuses. In the first case, the lines were transfected with the pTetOne plasmid carrying EGFR vIII, the wild-type variant (*EGFR*wt) was overexpressed in the second, and the expression of kinase-deficient EGFR was induced in the third. The above cell lines were tested under hypoxic conditions, as a result of which it was shown that with mutant EGFR vIII, as well as with stimulation of cells overexpressing EGFR with the help of growth factors, glioma cells become more vulnerable, which leads to their death [134].

One of the strategies for creating lines with glioma cell mutation patterns is the CRISPR/Cas9 technology, which allows comprehensive genome editing. This technology can be used to introduce loss-of-function (LOF) mutations that cause a frameshift or premature stop codon, resulting in gene knockout. It can also be applied for the introduction of reporter genes and epitope tags (knock-in), representation of single nucleotide polymorphisms (SNPs), and activation or repression of gene transcription by targeting promoter regions [135]. With this technology, it is possible to perform effective genetic screening and identify mutations that play a major role in gliomagenesis, as well as to create models with a specific genotype for drug testing.

Based on homologous recombination using the CRISPR/Cas9 system, a glioblastoma cell line was obtained from human brain organoids with inactivation of the *TP53* oncosuppressor and simultaneous expression of the HRasG12V oncogene [136]. This model suggests its possible use for fundamental studies of glioblastoma oncogenesis, since in 84% of patients with GB, the p53-ARF-MDM2 pathway is disrupted. This is primarily due to mutations in *TP53* and, as a result, the decrease in the activity of the tumor suppressor, which is accompanied by the acquisition of a more malignant phenotype [137]. The created cell line can be xenotransplanted into immunosuppressed mice to simulate the behavior of human glioblastoma in vivo and to conduct preclinical studies aimed at finding antitumor drugs [136].

In their research, Han et al. used temozolomide-resistant glioma cells and applied the CRISPR/Cas9 system to knock out the *ATRX* gene. This approach allowed them to experimentally determine the role of *ATRX* in the development of chemoresistance both in vitro and in vivo [138].

On the other hand, CRISPR/Cas9 has made it possible to label transcription factors to track their expression levels, stability, localization in the cell, and interaction with other proteins. Edited cell lines visualizing 60 different transcription factors were obtained based on human glioma stem cells using knock-in of epitope tags [139]. Such models will shed light on the mechanism of interaction of an antitumor agent with its therapeutic target.

As mentioned above, another significant oncogenic mutation R132H in the *IDH1* gene leads to the accumulation of the metabolite D-2-hydroxyglutarate (see the Introduction). A lot of modern research is devoted to studying the impact of this mutation, as well as the search for therapy against *IDH1*-mutant types of gliomas.

By introducing a single nucleotide substitution using CRISPR/Cas9, the astroglial cell line SVG with the heterozygous *IDH1* R132H/wt mutation and inactivated p53 was obtained [140]. The functionality of the model was confirmed by the detection of 2-HG in the medium, detection of changes in the DNA methylation pattern, and analysis of changes in gene expression. It was found that 315 genes related to cell migration were upregulated, while 538 genes involved in cell proliferation, differentiation (*PDGFRB*, *BHLH*), and ECM formation were downregulated. Inhibition of *IDH* revealed that expression profiles are regulated via the accumulation of oncometabolites, leading to altered methylation patterns. 2-HG promoted cell migration and invasiveness while suppressing proliferation, partly due to dysregulation of the MAPK, Notch, and Wnt pathways and downregulation of YAP expression.

A genetically modified mouse model with *IDH1* R132H mutation and PDGF overexpression was proposed, reflecting the phenotypic features of human gliomas, including 2-HG production, hypermethylation, and differential expression of a number of genes in comparison with a tumor without mutation in *IDH1* [141]. Methodically, the creation of the model included transfection of DF-1 cells using plasmids based on the RVAS/TVA system and subsequent insertion of cells into the cerebral cortex of mice for tumor development. The authors characterized major metabolic changes, including reduced glutamine and glutamate levels (due to 2-HG synthesis), dysregulation of Asp, NAA, and GSH, and increased gluconeogenesis.

By transfection, human oligodendroglioma cell lines with the expression of the oncogene *IDH1* R132H were constructed, which served as a screening tool for compounds directed at the *IDH* pathway [142]. 546 antitumor drugs and inhibitory compounds were tested on this model, among which were found those that specifically reduced the viability of mutant cells relative to cells with the wild-type *IDH1* variant. At the same time, some of them disrupted the synthesis of pyrimidine nucleotides de novo, namely BAY 2402234, inhibiting DHODH and some others. These compounds were also tested on a genetically modified mouse model of astrocytoma and demonstrated regression and suppression of tumor growth, due to which it was found that gliomas with a mutation in *IDH1* are vulnerable to compounds affecting the metabolism of pyrimidine nucleotides, disrupting the nucleotide balance and contributing to a decrease in DNA repair.

To obtain the LGG model, a group of scientists conducted a series of crosses of edited mice, after which they performed knockout of the *Atrx* gene using CRISPR/Cas9. Thus a combination of three driver mutations (*Idh1*, *Trp53*, and *Atrx*) was obtained, which led to the development of brain tumors [143]. The created model showed histological signs and molecular markers characteristic of human gliomas, demonstrating the expression of OLIG2, NESTIN, GFAP, and PDGFRA. This model allowed determination of the role of the rs55705857 polymorphism in gliomagenesis: mice were crossed with two other mouse lines carrying various mutations at the rs55705857 locus, and then they were also transduced with the LV-sgAtrx-Cre lentivirus. Both obtained lines had extremely high rates of glioma development, which indicates increased penetrance relative to tumors induced by mutations in *Idh1*, *Trp53*, and *Atrx* knockout, in which a long latency period was observed. Summing up, the researchers obtained a reliable model that can be applied in preclinical studies of therapeutic agents.

As genome editing and modification technologies developed, models became more complex and improved, and today many research groups create genetically designed model cell lines and animals to analyze the metabolic effects of mutations in *IDH1* and search for *IDH*-directed therapy [144,145,146,147,148].

Another approach to the development of genetically modified models is the use of viral particles to deliver transgenic constructs. Recombinant lentiviruses are primarily used to determine the significance of various genes and molecular pathways in the development of gliomas by delivering non-coding RNAs to cells that suppress the expression of the target protein [149].

Wang et al. demonstrated that overexpression of PDIA3P1 enhances glioma cell migration and invasion, promoting a transition to the highly invasive mesenchymal subtype under hypoxia. Lentiviral knockdown produced opposite effects [150]. They also showed that PDIA3P1 mediates activation of the NF-κB pathway. Additionally, their research revealed that hypoxia-induced HIF transcription factor upregulates PDIA3P1 expression.

Lentiviral particles can be used to develop models on cell cultures or animals. Researchers used lentiviral particles to create U87MG glioma cells expressing Gaussia luciferase and cyan fluorescent protein for drug screening. Testing of 1040 drugs revealed that cardiac glycosides induced glioblastoma cell apoptosis via TRAIL sensitizing [144,151,152].

The advancement of genetically modified models, particularly through the application of CRISPR/Cas9 technology and viral vector systems, has significantly enhanced the understanding of the molecular mechanisms underlying glioma pathogenesis and treatment resistance. The ability to create precise genetic alterations allows for the exploration of key oncogenic mutations and tumor suppressor gene inactivation, facilitating the study of their roles in gliomagenesis and therapeutic responses.

### 5.2. Models Based on Neural and Induced Pluripotent Stem Cells

The recent advances in cell modeling techniques, particularly the use of induced human pluripotent stem cells (iPSCs) and neural stem cells (NSCs), have provided valuable insights into the genetic alterations that lead to the development of glioblastoma [153].

A group of scientists at the University of Minnesota Medical School created glioblastoma cell models by introducing various combinations of genetic alterations that characterize different molecular subtypes of glioblastoma into human induced pluripotent stem cells [154]. To this end, iPSCs carrying CRISPR/Cas9-induced alterations of *PTEN/NF1* and *TP53/PDGFRA*, which are commonly observed in mesenchymal and proneural molecular subtypes of glioblastoma [155,156], were differentiated into neural progenitor cells (NPCs). The genetically engineered NPCs were then transplanted into immunocompromised animals, resulting in glioblastoma-like tumors. Histological studies confirmed that the tumors exhibited characteristics of glioblastoma. This study demonstrated that introducing different combinations of genetic driver alterations into cells on an isogenic background results in tumor models with distinct phenotypes. In addition, single-cell RNA sequencing analysis showed that these engineered models exhibited inter- and intratumor heterogeneity similar to that observed in glioblastoma tissue [157]. Importantly, these models were also suitable for drug sensitivity testing and longitudinal tumor evolution assessment.

In addition to iPSC-derived models, another significant glioma model was generated from human differentiated NSCs by knockout of the tumor suppressor gene *PTEN* via TALEN-mediated homologous recombination. In vitro experiments demonstrated that PTEN deficiency confers neoplastic potential on NSCs. In vivo experiments demonstrated that when introducing *PTEN*−/− cells into mouse models, NSCs formed neoplastic lesions and exhibited sensitivity to mitomycin C [158].

Moreover, another cell line was obtained from NSCs by infection of Myc-immortalized human NSCs with lentiviruses encoding dominant-negative p53 (p53DN) and/or a constitutively active myristoylated form of AKT (myr-AKT) [159]. Different experiments demonstrated high-level expression of the CHI3L1 gene by these cells and their derivatives [160]. CHI3L1 is a secreted glycoprotein with chitin-binding capacity, but lacking chitinase activity [161], that plays a role in tissue remodeling, inflammation, and cancer progression [162]. Also, CHI3L1 is highly expressed and associated with a poor clinical outcome in glioblastoma patients [163]. Results obtained by Chen et al. show that the relationship between CHI3L1 and PI3K/AKT/mTOR pathways has a positive feedback loop [160]. They describe that knockdown of the *CHI3L1* gene can significantly suppress tumor growth in mouse glioma models, highlighting its potential as a therapeutic target.

iPSCs and NSCs have significantly improved the study of genetic alterations in glioblastoma, highlighting the roles of PTEN deficiency and CHI3L1 expression in tumor development and aiding in therapeutic strategy formulation.

### 5.3. Models of Glioma Progression Driven by Hypoxia and HIF-1

Since hypoxia and hypoxia-induced signaling cascades play a central role in tumor development, they lead to more aggressive forms that are resistant to treatment. Multiple models are aimed at visualizing and evaluating the activity of HIF-1 (hypoxia-inducible factor 1) [22]. Models have been created using the luciferase reporter gene under the control of a promoter containing hypoxia response elements (HRE), using both mouse GL261 glioma cells [164] and the human U251MG glioma cell line to evaluate new approaches to tumor therapy [165].

Moroz and colleagues used lentiviral transduction to create U87MG glioma lines to visualize subcellular localization and stability of HIF-1 due to fusion with the luciferase reporter gene. The modified cells were xenotransplanted to create a mouse model, which was proposed by the authors to conduct preclinical studies and search for drugs that affect the activity of HIF-1 [166].

Wu et al. studied the invasive and migratory potential of U87MG and U251MG cells under hypoxia. Overexpression of HIF-1 in glioma cells increased H19 oncogene expression, which promotes angiogenesis and chemoresistance. The authors identified two specific HIF-1 binding sites in the *H19* promoter using luciferase reporters. HIF-1-mediated H19 expression stabilizes β-catenin, enhancing cell motility. This suggests a new potential therapeutic target for glioma treatment [167,168].

U3084MG cell lines were created that stably express variants of delta-like non-canonical Notch ligand 1 (DLK1). The expression of this protein enhances cell migration and invasion and is also associated with the acquisition of a stem phenotype by glioma cells. The cleavage of DLK1 during hypoxia depends on the activity of ADAM17 and HIF-1. The cells showed higher levels of the stem markers NANOG, OCT4, and SOX2, and their expression depended on the level of hypoxia [169]. These lines showed the association of DLK1 cleavage with the p53 and PI3K/AKT signaling pathways activation. DLK1 expression led to a more invasive pattern of glioma development in the mouse model. The authors suggest that the above events can be influenced therapeutically.

Models have been developed to mimic hypoxia conditions, which show the induction of epithelial–mesenchymal transition of human glioma cells. This process is mediated by the activation of HIF-1 and enhanced transcription of its target Gli1. Gli1 is a transcription factor and one of the participants in the Hedgehog signaling pathway, and its activation in gliomas contributes to the invasive cell phenotype and disease progression [159].

Some researchers focus on the extreme importance of the cellular components of the tumor microenvironment in its growth and development of hypoxia context [170]. On glioblastoma GL261 cells modified with a fluorescent UnaG reporter under the control of HRE (hypoxia response elements) transplanted into mice, it was possible to track the appearance of hypoxic zones and activation of HIF-1, as well as the attraction and migration of tumor-associated myeloid cells and cytotoxic T lymphocytes (CTL) into hypoxic niches for their reprogramming [171]. With the progression of GB and changes in the vascular network, a change in the pattern of distribution of tumor-associated myeloid cells and the formation of pseudopalisades were observed. When targeting hypoxia zones with the drug ifosfamide, activated at a low oxygen content, a significant decrease in the expression of the reporter system was observed, which confirms the therapeutic potential of drugs controlling the population of tumor cells in hypoxia.

In the Pantazopoulou et al. study, genetically engineered mouse glioma cells and human astrocytes were used to determine the contribution of hypoxic astrocytes to the tumor microenvironment. Activation of HIF-2a and its targets TGF-β1, IL-3, ANG, VEGF-A, and IL-1 alpha was detected, as well as an increase in tumor cell proliferation, maintenance of their stemness, and drug rejection due to the participation of hypoxic astrocytes in the formation of the extracellular matrix [172,173].

The complex relationship between hypoxia and tumor progression, especially in gliomas, highlights the critical role of HIF-1 in aggressive cancer phenotypes and therapy resistance. Several innovative cell models using reporter systems have been developed to visualize HIF-1 activity and study hypoxia-affected pathways, contributing to the identification of new therapeutic targets and oncogenic signaling in glioma invasiveness. Advanced genetic models have also clarified the dynamics of hypoxic zones and the reprogramming of immune cells, revealing the diversity of TAM subpopulations and the complexity of the immune landscape in glioblastoma.

### 5.4. Models Based on Cancer Stem Cells

Different models are proposed to study the role of glioma stem cells (GSCs) in tumor progression and the occurrence of recurrent conditions. GSCs are a subpopulation of self-renewing cells that are involved in tumor maintenance, which often leads to recurrence after treatment [174]. In addition, GSCs provide tumor heterogeneity and plasticity and are resistant to therapy due to enhanced DNA repair, production of reactive oxygen species, dormancy, and localization of cells in hypoxic niches [175].

In the study by Lee et al., GSC-specific reporter systems were developed using promoters—*SOX2*, *Oct4*, *Nanog*, and *CD133*—cloned into lentiviral constructs with RFP [176]. In response to chemotherapy, transduced non-glioblastoma stem cells began to show signs of a stem phenotype. However, this model had a few limitations: the reporter system was activated only after the formation of neurospheres, and there was no statistically significant difference when analyzing stem markers between RFP+ and RFP- populations. In addition, due to the knockdown of HIF1a, the GSC population decreased significantly, which indicates the importance of HIF1a in the acquisition of glioblastoma cells of an undifferentiated phenotype and further maintenance of the GSC population [176,177,178]. In GSCs that expressed the mutant variant of the *IDH1* R132H gene after lentiviral transduction, the study confirmed a decrease in β-catenin activity and inhibition of Wnt/β-catenin signaling. In addition, the cells showed a decrease in proliferation, migration, and invasion, accompanied by an increase in apoptosis and increased differentiation of GSCs [179].

Currently, many studies have confirmed that cancer stem cells are a limiting factor in glioma therapy, reducing its effectiveness [180,181,182]. Moreover, therapeutic stress provokes the transition of cells into a stem state. The created models provide opportunities to study the mechanisms of control of the GSCs population and the influence of the molecular profile on their phenotype and to find a treatment strategy that would consider the characteristics of GSCs and reduce the recurrent potential of the tumor.

### 5.5. Models for Studying Chromosomal Instability Influence

CIN has recently been identified as one of the most important factors in glioma pathogenesis and progression [183,184,185,186,187]. Cell models with induced chromosomal instability by gene modification or other methods, as well as allowing the assessment of the level of chromosomal instability, could be of great help in the search for personalized therapies [183,184,188].

The formation of oncogenic fusion proteins as a result of chromosomal translocations has been identified as a pivotal event in the development of human cancer [189,190]. Devendra et al. predicted and confirmed five in-frame fusions in patient samples. Subsequent analysis of data from the Cancer Genome Atlas revealed that the FGFR-TACC fusion was the most relevant to study. Then, they obtained cell lines by lentiviral transduction of FGFR-TACC3 fusion into Rat1 fibroblasts and the Ink4A/Arf−/− astrocytes to ascertain the role of this fusion in oncogenesis and chromosomal instability. The resulting cells exhibited comparable levels of fusion protein expression relative to those observed in patients with glioblastoma. Notably, the transduced astrocytes were subcutaneously injected into immunodeficient mice, which resulted in the formation of tumors. Furthermore, it was demonstrated that cells expressing FGFR-TACC fusions exhibited three to five times more errors in chromosomal segregation compared to control cells [191].

*BIRC5*, which belongs to the inhibitor of apoptosis (IAP) gene family, is highly expressed in tumors. It has been postulated that BIRC5 plays a role in carcinogenesis by inhibiting apoptosis, thereby increasing the survival of tumor cells [192,193]. The overexpression of BIRC5 by lentiviral transduction led to a notable elevation in the proportion of cells exhibiting mitotic defects and DNA damage in U87MG *TP53* WT and U251MG *TP53* R273H cells. An increase in the level of γH2AX and the number of pDNA-PKcs foci, which are markers of DNA damage, was observed in all *BIRC5*-transduced cell lines. Additionally, the study observed an increase in the number of structural chromosomal aberrations and aneuploidy in U251MG cells following transduction [194].

CIN has been identified as a critical factor in the pathogenesis and progression of gliomas. This is evidenced by recent studies, which have demonstrated its role in the formation of oncogenic proteins and increased incidence of chromosomal abnormalities. Cell models with induced CIN have demonstrated a significant increase in errors in chromosomal segregation and aneuploidy, indicating their potential for investigating mechanisms of carcinogenesis and developing personalized therapies.

## 6. Epigenetically Modified Glioma Cell-Based Models

Epigenetic changes are part of gliomagenesis [195,196], and there is an increasing need to test the new drugs targeted on epigenetic modifications.

As mentioned earlier, gliomas are more sensitive to temozolomide chemotherapy due to *MGMT* gene methylation, which is driven by mutations in the *IDH1* gene. Despite the understanding of the importance of *MGMT* methylation effects, a cell model was lacking for a long time. But in 2023, several researchers performed silencing of the *MGMT* by CRISPRoff-based targeted hypermethylation of the promoter [197,198]. CRISPRoff is based on the transcriptional repression domain of the Krüppel-associated boxed domain (KRAB). KRAB recruits TRIM28/HP1α, which enhances the stability of DNA hypermethylation to provide a suitable tool for gene silencing [199,200,201]. A further study used the dCas9/DNMT3A catalytic domain [202,203] to target the *MGMT* promoter methylation and its enhancers. dCas9/DNMT3A is based on a single RNA that is required to direct the fusion proteins of the nuclease-deficient Cas9 with DNMT3A (DNA (cytosine-5)-methyltransferase 3A) to target sites in the genome. Both approaches are transient, but the CRISPRoff-based approach is more durable [199,200,201,204] (Figure 3).

There is another attempt to create a model that demonstrates the influence of mutation in *IDH1* on DNA hypermethylation [205]. Moore et al. performed a knockout of *IDH1* R132H by pSpCas9(BB)-2A-GFP (PX458) construct in patient-derived glioma cells [145,206]. This widely demonstrates changes in the methylation profile of glioma cells in contrast to the reference cell line. Several studies utilized acetylated p53 to investigate the molecular mechanisms underlying gliomas and to facilitate drug screening [207].

TERT has a crucial role in gliomagenesis. Taghavi rad et al. demonstrate that suppression of TERT by CRISPR/Cas9 in glioma cells may result in increasing sensitivity to chemotherapeutic drugs [208].

Another approach to creating cell models of epigenetic changes is the utilization of microRNA expression [209]. miR-16 was overexpressed in glioma cell lines, which resulted in proliferation and migration decrease [210]. Knockdown of miR-21 enabled the generation of a cell line with low proliferation ability [211].

However, despite the existence of epigenetic research, some aspects are still unexplored, which prevents the creation of a complete and stable epigenetic cell-based model.

## 7. 3D Cell-Based Constructs

The tumor microenvironment is one of the main factors determining the behavior of the tumor cells and their response to therapy. Close contact with the extracellular matrix (ECM) components, cancer, and stromal cells promotes tumor development and supports the population of cancer stem cells, proliferation, and infiltrative growth. Deficiency of nutrients and oxygen due to insufficient vascularization and limited transport of molecules because of the presence of the blood–brain barrier is also important. Two-dimensional cell cultures do not allow mimicking of the range of conditions existing in a tumor. This means that the results of studies conducted using such models may not be representative [112]. The most promising approach for studying the mechanisms of carcinogenesis, assessing the invasive potential, and drug screening is three-dimensional cell culturing (Figure 4). Such models fill the gap between 2D cultures and in vivo models and allow researchers to reduce the time to evaluate the effectiveness of potential therapeutics.

### 7.1. Spheroids

The simplest form of three-dimensional culturing is spheroid culture. Spheroids are rounded cell aggregates. They could be formed using standard approaches: the hanging drop method, culturing on low-adhesion coatings, culturing in serum-free media, cell culture rotation, and magnetic levitation [212]. The first three methods are the most popular for creating spheroids from glioma cells [213,214,215].

Spheroid cultures mimic close cellular contact, which facilitates paracrine regulation. Dense packing of the cells into a spheroid allows recreation of the reduced diffusion of nutrients and hypoxia [216,217] existing in real tumors due to the lack of vascularization. Hypoxia is a key factor in the pathogenesis of gliomas, which enhances tumor development by increasing the expression of stemness (CD133, SOX-2) and drug resistance markers (TIMP-1, Lamp-1) in tumor cells [218].

Glioma cell lines and patient-derived cultures can be used as sources of cells for spheroid formation. The advantage of spheroid cultures made of conventional cell lines is their simplicity and reproducibility [219,220]. These lines have a high growth rate, do not require special media and supplements for cultivation, and allow spheroids to be generated using standard protocols without modifications. Cells of the permanent glioma lines in spheroids change their molecular profile, increasing the expression of tumor stem cell markers, ECM components, some chemokines, and molecules and mediating interaction with immune cells [221].

The main disadvantage of spheroids based on permanent cell lines is abnormally low heterogeneity due to prolonged cultivation [222]. It is known that gliomas have significant intratumor heterogeneity and contain many different cell types, each of which has a specific function in the development and progression of gliomas, as well as in drug resistance [223]. Cell-line-derived spheroids do not allow mimicking of the entire spectrum of the intercellular interactions and are a useful but limited tumor model. This problem can be partially solved by using low-passage patient-derived cultures to create spheroids [224] that keep the heterogeneity of the tumor.

Screening of drugs [225,226,227] and other therapeutic agents [84,228,229] using spheroid models for the development of anti-glioma therapies has become a routine practice for researchers.

### 7.2. Organoids

A more complex three-dimensional model of gliomas is an organoid. This is a cellular aggregate consisting of different cell types, which allows it to imitate the microenvironment that exists in tumors in vivo.

Several approaches have been described for the formation of glioma organoids. Organoids can be generated from patient-derived tumor tissue samples that can be mechanically and enzymatically dissociated to obtain a cell suspension [107,230] or cut into pieces of approximately 1 mm in size [119,231,232]. The organoids are then formed using the required number of cells or tumor fragments by the spheroid formation methods mentioned above. There are reports of successful cultivation of glioma organoids from patient-derived material for over a year [107].

The cells of organoids formed this way generally retain the heterogeneity and morphology of the parent tumor and replicate its expression profiles and mutation patterns [119]. Organoid cultures demonstrate a gradient of cancer stem cell (CSC) density. Their number is inversely proportional to the level of hypoxia in the center of the organoid. Thus, analogs of two zones characteristic of a tumor are formed: perivascular and hypoxic niches. Glioma stem cells on the periphery are actively proliferating, while single GSCs of the hypoxic core are dormant [107].

Interesting data were obtained when comparing two methods of organoid formation: from a tumor cell suspension with Matrigel^TM^, and from pieces of tumor tissue. Organoids formed by the latter demonstrate greater similarity to the parent tumor in terms of genotype, expression of SOX-2, NESTIN, and GFAP, proliferation index, and formation of a hypoxic core [233].

During long-term cultivation of glioma organoids obtained from a piece of tumor in Matrigel^TM^, some of the cells are able to invade the matrix and form spheroids. Such spheroids have reduced cellular heterogeneity compared to the parent tumor and are apparently formed mainly from malignant cells [232].

Most cell-based models are derived from high-grade glioma cells. This happens because low-grade glioma cells are difficult to maintain in culture [234]. They have a low proliferation rate and poorly adapt to in vitro conditions or lose their unique phenotype in culture. However, there are reports of successful preparation of low-grade glioma organoid cultures that retain the cytoarchitecture of the parent tumor for 6 months of cultivation [231]. Perhaps organoid cultures are the most convenient model for low-grade gliomas.

Another approach to creating 3D models based on organoids is to graft tumor cells into cerebral organoids from embryonic stem cells or induced pluripotent cells. The resulting 3D culture is called a glioma cerebral organoid (GLICO). Cerebral organoids are obtained by culturing cells on low-adhesion plates. The resulting aggregates are embedded in Matrigel^TM^. The organoids are incubated with a suspension of glioma cells or isolated GSCs. Typically, 24 h is enough for cell grafting, and the cell engraftment efficiency reaches 100%. A tumor formed this way exhibits an infiltrative growth pattern and retains the main morphological and genetic characteristics of the parent tumor used as a source of cells [235,236,237]. GLICO models keep key characteristics over 4 months of cultivation [235].

It has been described that culturing standard cell lines in GLICO models also leads to infiltration of the cerebral organoid by tumor cells and a change in their expression profile with the formation of a phenotype characteristic of tumors in vivo [238].

Genetic engineering techniques are used to create 3D models of gliomas based on cerebral organoids. It has been described that *RAS* activation in combination with *TP53* deletion in some cells of the cerebral organoid leads to the formation of an invasive tumor with expression profiles characteristic of glioblastoma [136]. A similar study describes the development of a model based on a cerebral organoid with *MYC* overexpression and *TP53* loss [239].

Organoid cultures are already used to search for therapeutic targets and study the effectiveness of anti-glioma drugs based on a personalized approach [240,241,242]. The importance of the microenvironment in the formation of drug resistance has been confirmed in GLICO models and in organoids from patient samples. Organoid cultures are more resistant to the effects of therapeutic agents than two-dimensional cell cultures [235]. This may be due to the sustained viability of the pool of resistant CSCs [107].

Some researchers also describe the use of glioma slice culture for drug screening. Such cultures retain the morphology of the parent tumor for only 2 weeks [243,244], which is their disadvantage in comparison with organoids.

Spheroids and organoids are used to create more complex cell-based models by matrix embedding, implantation into brain slices, or xenotransplantation for in vivo research.

### 7.3. Scaffolds and 3D Bioprinting

It is known that cell transformation is accompanied by a change in the composition and mechanical properties of the extracellular matrix, which contributes to tumor progression, drug resistance, and the formation of a physical barrier for the delivery of a therapeutic agent to the pathological focus [245,246,247]. In this context, three-dimensional models based on cells and biomaterials that mimic intercell interaction and the extracellular matrix are of particular interest for the search and selection of therapeutics. The various biomaterials are used as substrates for cell culturing on its surface or as a three-dimensional environment for cell embedding.

Synthetic and natural biopolymers can be used alone or in combination to create scaffolds that mimic ECM [248]. It should be noted that synthetic polymers are biologically inert and require additional modifications. Moreover, these compounds can degrade to form cytotoxic substances or cause immune reactions. Thus, biopolymers are more attractive for researchers [249].

Often, the composition of the biomaterials for tissue-engineered structures in 3D modeling of gliomas includes hyaluronic acid (HA), which is one of the main components of the brain ECM and plays an important role in tumor progression. Three-dimensional cultures of gliomas with HA support a pool of tumor stem cells [250,251]. Matrix based on hyaluronic acid and chitosan also promotes the expression of epithelial–mesenchymal transition genes (*CD44*, *HIF-1α*, *SNAI1*, *TWIST*, *STAT3*, etc.) and genes regulating migration and adhesion (*MRTK*, *AXL*, *MUC1*, *EPCAM*, etc.) [252,253].

Not only carbohydrates but also ECM proteins are used as components of hydrogels. The addition of proteins leads to the activation of cell migration into the hydrogel matrix, promoting stemness and resistance to therapy [254]. Cells of glioma patients cultured in collagen I and IV matrix have a high migration rate [255] and demonstrate patterns of treatment resistance similar to the parent tumor [256]. A similar effect has been described for cell lines [257]. Cells cultured in composite materials with HA and Arg-Gly-Asp peptide (RGD) demonstrated resistance to the action of alkylating agent, caused by the activation of signaling through CD44 and integrin αV [258].

Gelatin-methacrylate (GelMA) gels have become widely used in glioma tissue engineering due to their tunable mechanical properties. Cells cultured in such scaffolds demonstrate high invasive potential and resistance to alkylating agents and acquire a mesenchymal-like phenotype corresponding to cells of the hypoxic and perivascular niches [259]. GB models based on GelMA hydrogels with HA have shown the role of hyaluronan and CD44-mediated signaling in resistance to EGFR inhibitor therapy [260,261] and maintenance of the stem cell pool.

Matrigel^TM^ and decellularized matrix are actively used as a basis for cell-based models, allowing mimicking of the entire complexity of the ECM composition. However, the disadvantage of such scaffolds is low reproducibility from batch to batch. Matrigel^TM^ is often used as a substrate for organoid formation [107] or as a scaffold for invasion models [220,262]. Culturing glioblastoma cells in a decellularized brain matrix significantly alters cell morphology, invasion dynamics, and the gene expression profile. In particular, there is activation of integrin signaling, a shift in cellular metabolism toward anaerobic glycolysis, activation of genes associated with the development of drug resistance, and maintenance of GSC stemness [263,264].

An important aspect in glioma modeling is not only the matrix composition but also its mechanical properties. Increased matrix stiffness (up to 26 kPa) leads to activation of migration, proliferation, and matrix remodeling genes in U87MG cells [265]. Importantly, cell lines and patient-derived cells respond to stiffness differently. There is evidence that patient-derived cells proliferate better in soft scaffolds (up to 40 Pa) [266], but increased stiffness leads to activation of HIF1α, VEGFa, and MMP2 and drug resistance [267,268].

Recently, bioprinting has been used to create tissue-engineered constructs based on biomaterials. This approach allows the development of multilayer models with several types of cells and scaffolds that replicate the structural complexity of tissues. Biopolymers such as hyaluronic acid, chitosan, alginate, collagen, and gelatin could be easily gelated in physiological conditions and are most often used for printing the tissue-engineered structures [269]. Most bioprinted cultures recreate two zones: the tumor zone and the zone of normal brain tissue around it [270]. For example, a three-dimensional model of glioblastoma was created with different zones of scaffold rigidity corresponding to normal brain, tumor, and endothelial tissues [271]. In another case, bilayer tetracultures consisting of GSCs, macrophages, astrocytes, and NSCs mimicking tumor and surrounding normal tissue more accurately replicated the transcriptome profile of GB than spheroid cultures and demonstrated the role of immune cells in tumor progression [272].

### 7.4. Blood–Brain Barrier Models and Microfluidics

One of the important problems in the treatment of gliomas is the delivery of therapeutic agents to the pathological focus. The blood–brain barrier (BBB) is a complex structure that limits the transport of substances from the blood to the brain. The cellular component of the BBB includes endothelial cells, pericytes, and astrocytes, and the extracellular component is formed by a network of proteins (collagens, laminins, and nidogens) and some proteoglycans. The barrier function of the BBB is provided mainly by tight junctions mediated by both cells and the ECM [273,274].

Despite the massive disruption of the BBB caused by tumor development, it remains intact in the tumor bed. Cells in this area remain protected from all of the large-molecule drugs and most small-molecule compounds and give rise to tumor recurrence [275]. In this context, blood–brain barrier models are of interest for the development of drugs delivered in a minimally invasive manner, as well as for studying the behavior of tumor cells in perivascular niches.

The simplest approach to creating a BBB model involves culturing a layer of epithelial or endothelial cells on the surface of a Transwell^TM^: two chambers separated by a BBB analog [276,277]. Bone marrow endothelial cells (BMECs) derived from induced pluripotent stem cells are also used to create a barrier in such systems [278,279]. There is evidence that SOX-18 expression plays an important role in supporting BMECs maturation and improves their barrier properties [280]. However, co-cultivation of BMECs and pericytes on the Transwell^TM^ surface does not improve the barrier properties of the system [281]. ECM proteins can be used to form the BBB scaffold in such systems [282,283].

The formation of tight junctions between cells in such models is confirmed immunohistochemically, while the integrity of the cell monolayer and the function of transport systems are assessed by the permeability of fluorescent dyes [276,277] or by measuring transendothelial electrical resistance [279]. The developed model is successfully used to evaluate the transport of cytotoxic drugs and promising candidates, for which the efficiency of passing through the BBB is unknown [276].

More complex approaches to creating BBB models are based on microfluidic systems. A microfluidic chip is a device with several chambers connected by a network of channels into a flow cell. Each of the chambers can be loaded with different types of cells with or without scaffolds, and the fluid flow between the chambers allows simulation of the vascular bed. The basis for creating such systems can be both original [284] and commercial chips [285]. To mimic the barrier, monocultures of endothelial cells, such as hCMEC, iBMEC, or HUVEC [284,286,287], or complex mixed cultures with the addition of astrocytes and pericytes [285] are used. To assess the behavior of tumor cells, patient-derived GSCs can be introduced into the model [284,287]. The most commonly used matrices are hydrogels made from ECM proteins: collagen types I and IV, laminins, fibronectin [285,287,288], and Matrigel^TM^ [284].

### 7.5. Co-Culture with Organotypic Brain Slices

Organotypic brain slice cultures are suitable platforms for co-culture with tumor cells, as they preserve viability and brain cytoarchitecture [289,290]. There are studies that describe successful development of a glioblastoma model based on organotypic cultures of mouse brain slices and single tumor cells [291], as well as spheroids and organoids [289,292]. In recent research, organotypic rat brain slices [293] and slices of normal human brain [294] were used to create similar models.

When implanted into different areas of the brain, the cells demonstrate different morphology. Thus, after microinjection into the lateral area of the corpus callosum, the cells of the spheroids from the GSCs acquire an infiltrating GFAP-positive low-proliferating phenotype compared to the spheroids implanted into the subependymal zone [290]. After implantation into the area between the cortex and the striatum, the organoids made of cells obtained from patients exhibit invasive properties in contrast to the spheroids of the standard cell line [292]. Conflicting data were obtained in a similar study of the invasion of spheroids from the permanent cell lines U87MG, U118MG, U251MG, LN-229, LN-319, T98G [289,295], and C6 [296].

Complex models based on organotypic brain slice cultures allow studies of the invasive potential of cells [295], the role of individual molecules in glioma progression [297,298], and the mechanisms of glioma immunoresistance [299] for the development of effective therapeutic strategies. Similar models are already being used to test small molecules that suppress cell invasion [289,300] and other anticancer agents [290,301,302].

### 7.6. In Vivo Cell-Based Glioma Models

The most advanced of all existing three-dimensional models, perhaps, is the tumor xenograft. Such models consider all the effects of the tumor environment, including the ECM and surrounding normal nerve and stromal cells. The use of patient-derived tumor organoids allows for the full replication of intratumoral heterogeneity and enables the implementation of personalized therapy strategies. Today, the most representative results should be expected from drug screening using these systems.

It has been established that transplantation of patient-derived glioma organoids (patient-derived orthotopic xenograft, PDOX) leads to the formation of a tumor in the mouse brain similar to the parental one, the cells of which reproduce patterns of expression and mutations [118] and actively (in 92% of cases) migrate to healthy tissue. Moreover, the transplanted cells had a high affinity for white matter [119]. There are successful cases of testing the effectiveness of anti-glioma drugs [303] and radiotherapy [304] on PDOX models.

Successful grafting of not only patient-derived xenografts but also glioma cultures based on cerebral organoids has been demonstrated [136].

When designing such in vivo models, some features must be taken into account. When comparing the behavior of standard U87MG cells and patient-derived GSCs in vivo, it was found that U87MG cells formed a solid tumor and led to the appearance of neurological symptoms in mice after 19 days, while GSC tumors had an infiltrative growth pattern and later clinical manifestations [292]. It is also important to note that not only the cell source but also the method of cultivation prior to intracranial transplantation plays a significant role. Cells cultured in 3D conditions exhibit better growth compared to those cultured in 2D [305]. However, the success of grafting is not affected by the presence of non-tumor cells in the graft [118].

Creating xenograft models from low-grade gliomas is a challenge. It has been shown that low-grade glioma organoids do not progress for 18 months when transplanted into mice [306].

Not only orthotopic but also subcutaneous transplantations of tumors to mice are described. Despite their limitations related to localization, this approach has been established as a faster and simpler yet effective method for selecting personalized therapy [230].

## 8. Future Models That Should Be Developed

It should be clarified that all those types of models serve different purposes. Genetically modified models can help to identify fundamentally new, previously unexplored molecules with antitumor activity. The search for drugs in personalized medicine involves the use of patient-derived low-passage cell cultures that retain the unique initial genotype of the person.

Now, there is a necessity to create models that would demonstrate a variety of genetic variations and expression profiles for the targeted search for drugs that would offset the effects of these mutations on the metabolism and other characteristics of tumor cells. Most of the existing genetically modified models are aimed at studying the mechanisms of gliomagenesis. For drug discovery, the most promising strategy is to use reporter systems that allow for rapid visual evaluation of cell status. For example, reporters visualizing the activation/inactivation of the signaling pathways [307,308], enzymatic activity [309], the action of the transcription factor [310], the functioning of the promoter, the formation of hypoxic zones [171], and so on.

The previously mentioned models visualizing hypoxia or related proteins can be improved by creating a pool of genetically modified lines based on cells carrying various driver mutations. This would help to identify the significance of these mutations in adaptation to hypoxia and further progression of gliomas, as well as to search for HIF-1/2 inhibitors and for other participants in the molecular cascade [311].

As previously stated, another crucial factor that impacts glioma pathogenesis and progression is CIN [183,184,185,186,187]. There is a necessity for the development of models that can estimate chromosomal instability levels in response to specific therapeutics. Such models may prove invaluable in the pursuit of personalized therapy.

Approaches to measuring chromosomal instability include fluorescent labeling of chromatin-related proteins, operator/reporter systems, and fluorescently labeled artificial chromosomes or gene-editing tools. All these methods maintain cell viability and proliferation [312,313].

The utilization of human artificial chromosomes (HACs) comprising a fluorescent reporter gene enables the evaluation of HAC copy number alterations through the application of fluorescence imaging techniques, such as flow cytometry or fluorescent microscopy [314,315,316,317,318]. Lee et al. employed GFP-expressing HACs in conjunction with flow cytometry to evaluate the rate of HAC loss (i.e., measuring chromosomal instability) in response to exposure to various chemotherapeutic drugs [319,320].

The impact of chemotherapeutic agents and driver mutations on CIN in glioma subtypes can be examined using HACs. Although no studies have focused on this, introducing HACs with unstable GFP and overexpressing polymorphic gene variants in glioma cells can create models to assess the effect of these variants on HAC loss. This approach helps evaluate the role of genetic variants in CIN progression and drug response, offering valuable prognostic and therapeutic insights. Another approach to using HACs in glioma research is to evaluate sensitivity to new therapeutics by monitoring shifts in CIN levels under their influence.

In the context of improving existing epigenetic models, it is necessary to develop specific and efficient editing systems that can provide long-term epigenetic changes.

To effectively model the complex interactions between tumor cells, the surrounding microenvironment, and the ECM, future 3D systems must accurately reflect the cellular composition and biochemical and mechanical properties of the ECM. Tissue engineering and bioprinting technologies have emerged as groundbreaking methodologies for constructing such 3D models. These approaches allow for precise control over scaffold architecture, cellular arrangement, and the incorporation of multiple cell types, thereby facilitating the recreation of the tumor microenvironment.

For personalized therapeutic approaches, models based on patient-derived cells are crucial. Patient-derived tumor organoids (PDTOs) have been established as a promising tool for simulating the unique characteristics of individual tumors, allowing for patient-tailored drug screening. PDTOs can retain the genetic and phenotypic diversity of the primary tumor, providing insights into the heterogeneous responses to therapies and aiding in the identification of potential treatment strategies.

To identify and evaluate new therapeutic strategies for glioma treatment, it is crucial to combine genetic and/or epigenetic modifications with advanced 3D cultivation techniques. This integrated approach not only enables the study of tumor heterogeneity and plasticity but also allows for the dynamic tracking of key cellular processes such as proliferation, invasion, apoptosis, and response to treatment. Furthermore, by closely replicating the complex architecture of the tumor and its microenvironment, this strategy enhances the predictive power of preclinical models, providing a more reliable platform for drug testing and personalized therapy development.

## 9. Conclusions

Existing glioma cell-based models have both advantages and limitations. On the one hand, they allow us to study the mechanisms of gliomagenesis, tumor–cell interactions with the microenvironment, and their response to various therapeutic agents in controlled settings. On the other hand, such models may not fully capture the complexity of gliomas, especially the dynamic interactions between tumor cells and the surrounding neural tissue, which may limit their prognostic value. Overall, although glioma cell-based models have significant potential to improve our understanding of tumor biology and enhance drug discovery processes, ongoing research is essential to improve these approaches and address their inherent limitations. By refining models, we can pave the way for more effective therapeutic strategies tailored to individual tumor characteristics for each patient and perform high throughput screening of new anti-glioma therapeutic agents.

## Figures and Tables

**Figure 1 cells-13-02085-f001:**
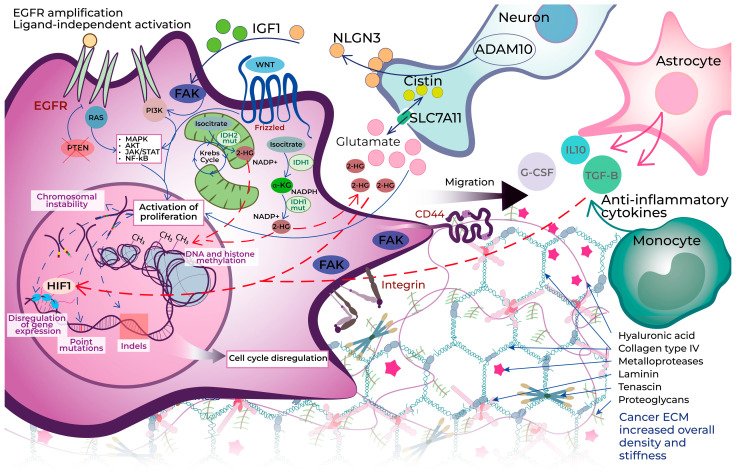
Key molecular pathways, onco-associated molecules, and the microenvironment involved in gliomagenesis as prospective targets for glioma therapy that should be modeled in vitro.

**Figure 2 cells-13-02085-f002:**
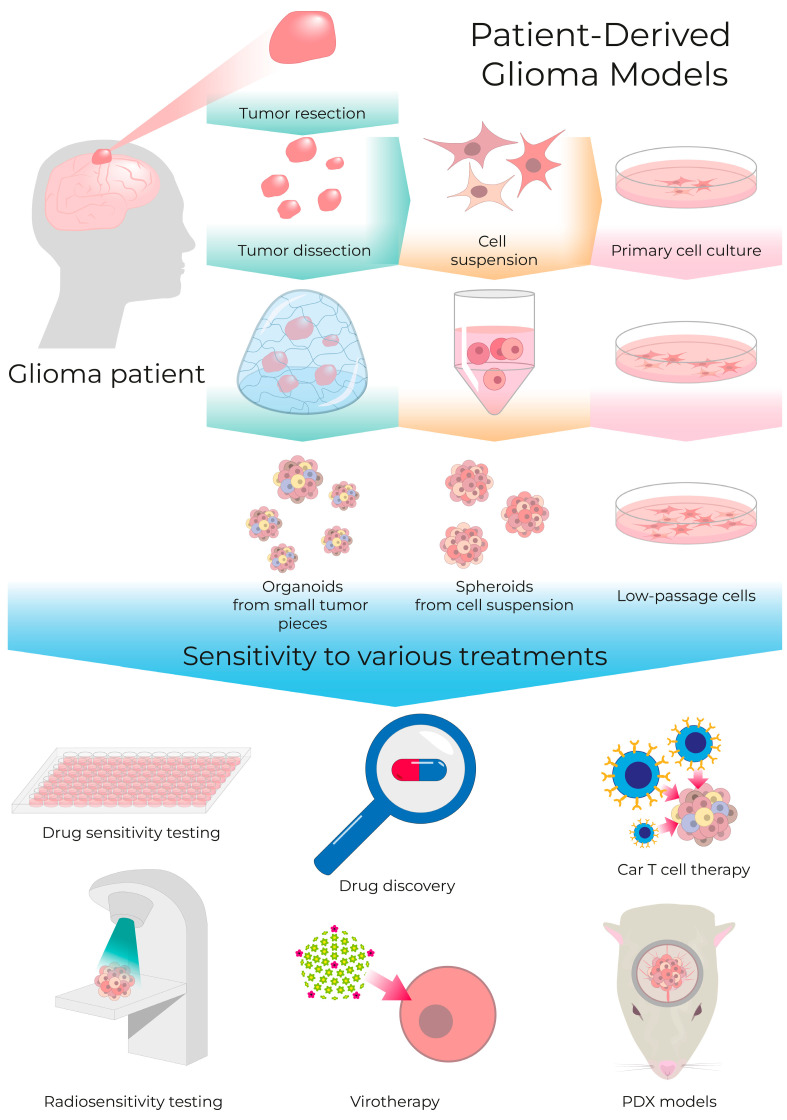
Application of the patient-derived glioma cell-based models.

**Figure 3 cells-13-02085-f003:**
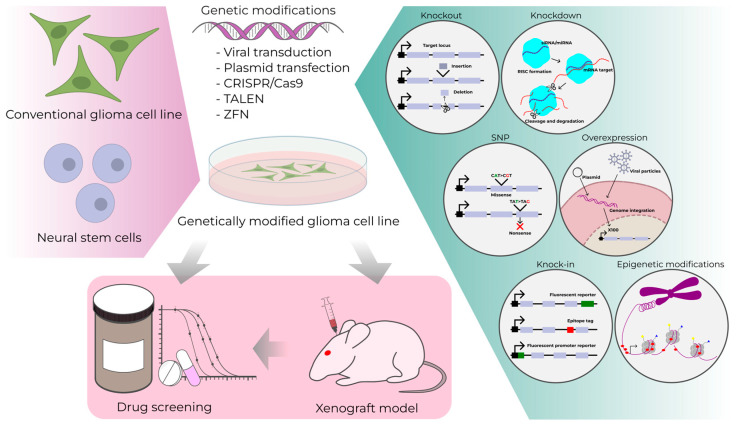
Approaches employed to create genetically modified glioma cell-based models and their subsequent applications. The left section shows the cell types most frequently utilized for modification, while the right section illustrates the various genetic and epigenetic modifications.

**Figure 4 cells-13-02085-f004:**
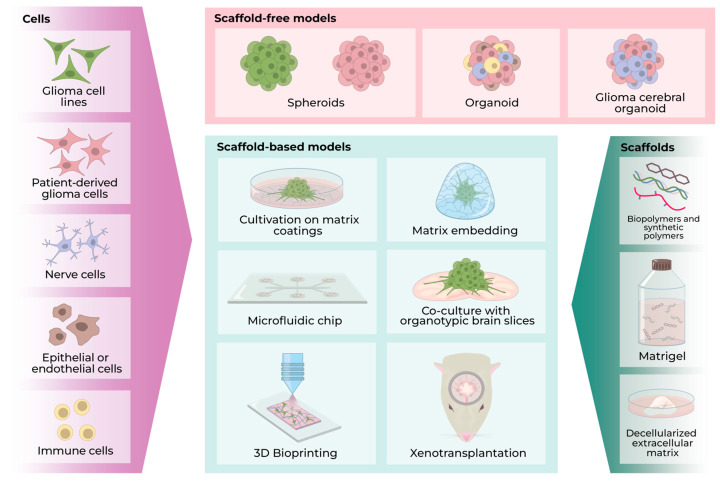
Approaches for creating 3D cell-based glioma models. The left side of the figure shows the main cell sources for scaffold-free and scaffold-based models, while the right side shows the scaffold options for scaffold-based models.

## Data Availability

Not applicable.

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
