# Peer review of "Cell-Based Glioma Models for Anticancer Drug Screening: From Conventional Adherent Cell Cultures to Tumor-Specific Three-Dimensional Constructs"

_cells, 2024, doi:10.3390/cells13242085_

Round 1
Reviewer 1 Report
Comments and Suggestions for Authors
The review article entitled “Cell-based glioma models for anticancer drug screening: from conventional adherent cell cultures to tumor-specific 3D constructs” submitted by Dr. Lanskikh and colleagues, extensively addresses the complexity of studying gliomas in the different experimental models already described. The authors managed to present the state of the art of in vitro studies of brain tumors in an objective, concise and pleasant manner.
Since the authors mention at various times the diversity of tumor microenvironments in the different types of gliomas, especially when the samples are from primary tumors, it would be important to mention single cell studies, since there is already a lot of information on this subject. These studies have even suggested new therapeutic targets and mechanisms of tumorigenesis. In the context of this review, this discussion should be unavoidable. Some examples are:
Karimi E, et al Single-cell spatial immune landscapes of primary and metastatic brain tumours. Nature. 2023 Feb;614(7948):555-563. doi: 10.1038/s41586-022-05680-3.
Xie Y, et al. Single-cell dissection of the human blood-brain barrier and glioma blood-tumor barrier. Neuron. 2024 Sep 25;112(18):3089-3105.e7. doi: 10.1016/j.neuron.2024.07.026.
Ochocka N, Segit P, Walentynowicz KA, Wojnicki K, Cyranowski S, Swatler J, Mieczkowski J, Kaminska B. Single-cell RNA sequencing reveals functional heterogeneity of glioma-associated brain macrophages. Nat Commun. 2021 Feb 19;12(1):1151. doi: 10.1038/s41467-021-21407-w.
Author Response
- Summary
Thank you very much for taking the time to review this manuscript. Thank you for your constructive comment. Please find the detailed responses below and the corrections in track changes in the re-submitted files.
- Point-by-point response to Comments and Suggestions for Authors
Comments 1:
Since the authors mention at various times the diversity of tumor microenvironments in the different types of gliomas, especially when the samples are from primary tumors, it would be important to mention single cell studies, since there is already a lot of information on this subject. These studies have even suggested new therapeutic targets and mechanisms of tumorigenesis. In the context of this review, this discussion should be unavoidable.
Response 1: Thank you for pointing this out. We completely agree with you regarding the importance of single-cell studies for a deeper understanding of glioma heterogeneity and the identification of new therapeutic targets.
In response to your suggestion, we have revised the manuscript to include a discussion of recent findings from single-cell technologies. We have highlighted several single-cell-based studies that revealed promising new glioma molecular signatures and demonstrated the functional heterogeneity of glioma and glioma-associated cells (see lines 185-193, 305).
- Additional clarifications
We believe these additions make the review more comprehensive and up-to-date, reflecting the latest advances in glioma research.
Thank you again for your valuable suggestions, which have helped improve our manuscript.
Reviewer 2 Report
Comments and Suggestions for Authors
This manuscript has two objectives: 1) Describe conventional adherent glioma cell lines, patient-derived models, genetically and epigenetically modified cell models, and 3D cell-based constructs models that can be used to evaluate the efficiency of new drugs, and 2) a review of marine-derived molecules that could be tested to target glioma cells. The second aim does not correspond to the title of the manuscript “Cell-based glioma models for anticancer drug screening...” and must be reported in a separate manuscript.
The introduction summarizes the different subgroups of gliomas and the associated genetic alterations. This section could be simplified and better designed to mark a link with the advantages and disadvantages of existing glioma cell-based models for anticancer drug screening.
Most sections are generic and lessen the reader's interest, as this information is widely known in the scientific community and does not contribute significantly to the overall objective of this review. Interesting sections describe tumor organoids, scaffolds, 3D bioprinting, and the transplantation of patient-derived glioma organoids into the murine brain, leading to human-like brain tumor formation.
Sub-sections 3.2 and 3.3 have the same title.
Author Response
- Summary
Thank you very much for taking the time to review this manuscript. We appreciate your detailed feedback on our manuscript. Please find the detailed responses below and the corrections in track changes in the re-submitted files.
- Point-by-point response to Comments and Suggestions for Authors
Comments 1:
This manuscript has two objectives: 1) Describe conventional adherent glioma cell lines, patient-derived models, genetically and epigenetically modified cell models, and 3D cell-based constructs models that can be used to evaluate the efficiency of new drugs, and 2) a review of marine-derived molecules that could be tested to target glioma cells. The second aim does not correspond to the title of the manuscript “Cell-based glioma models for anticancer drug screening...” and must be reported in a separate manuscript.
Response 1:
Thank you for your valuable feedback and for highlighting this point. We understand your concern regarding the second objective, which discusses marine-derived molecules. We have chosen to remove this chapter from the current review and reserve it for inclusion in a separate article. Thank you for bringing this to our attention.
Comments 2:
The introduction summarizes the different subgroups of gliomas and the associated genetic alterations. This section could be simplified and better designed to mark a link with the advantages and disadvantages of existing glioma cell-based models for anticancer drug screening.
Response 2:
You make a fair point regarding the introduction: it does come across as a bit overloaded with detail in some parts. In response to your feedback, we have restructured the manuscript as follows: the key molecular features of gliomas have been moved to a dedicated section, while the introduction has been streamlined to serve as a brief overview, introducing the main topics of the review (see lines 25-68). However, we believe that the information on the molecular features of gliomas is crucial and cannot be removed. Molecular characteristics are directly tied to the development and refinement of glioma models, as they inform the design of more accurate and clinically relevant models for drug testing. Understanding these features is essential for the proper modeling of gliomas and for evaluating the efficacy of potential therapies (see lines 94-105). Therefore, we consider this information integral to the manuscript's focus.
Comments 3:
Most sections are generic and lessen the reader's interest, as this information is widely known in the scientific community and does not contribute significantly to the overall objective of this review. Interesting sections describe tumor organoids, scaffolds, 3D bioprinting, and the transplantation of patient-derived glioma organoids into the murine brain, leading to human-like brain tumor formation.
Response 3:
Your observation about certain sections being generic is valid. However, our work stands out by covering all key types of cell-based glioma models, ranging from traditional 2D glioblastoma cell lines to genetically modified cells and advanced PDX models, offering a well-rounded perspective on the topic. Below, we address your concerns by emphasizing the strengths of our work in comparison to existing literature. To this end, we have performed a concise review of relevant articles to underscore our manuscript's unique contribution.
Many existing reviews on glioma models offer valuable insights but tend to focus on specific aspects of the field. For example, some concentrate on preclinical animal models [https://doi.org/10.3390/cells10030712, https://doi.org/10.1016/j.bbcan.2020.188458, https://doi.org/10.1111/cas.13351], while others explore organoid systems but give less attention to simpler cell models [https://doi.org/10.20892/j.issn.2095-3941.2023.0061]. Similarly, some reviews focus on in vitro approaches, leaving other strategies less explored [https://doi.org/10.1007/s13402-022-00684-7, https://doi.org/10.3934/genet.2018.2.91].
There are also reviews that highlight genetically engineered animals without delving deeply into genetically modified cell models [https://doi.org/10.1007/s00401-017-1671-4, https://doi.org/10.1111/cas.13351], or those that focus exclusively on patient-derived models [https://doi.org/10.3390/cells8101177]. While some articles cover a wide range of cell-based models, they often focus solely on glioblastoma, without addressing the unique characteristics of lower-grade gliomas [https://doi.org/10.3389/fonc.2020.614295, https://doi.org/10.1016/j.bbcan.2023.189059, https://doi.org/10.3390/cancers11010044, https://doi.org/10.1242/dmm.040386].
In our manuscript, we aim to build upon and complement these valuable contributions by providing a more comprehensive perspective. We integrate various modeling approaches across the glioma spectrum, addressing challenges like heterogeneity and drug delivery mechanisms with innovative tools such as scaffold-based constructs and bioprinting. Additionally, we suggest potential directions for developing future models, aiming to contribute to the ongoing effort to advance glioma research.
Comments 4:
Sub-sections 3.2 and 3.3 have the same title.
Response 4:
Thank you for pointing out the duplicate titles in sections 3.2 and 3.3. We will modify the titles (see lines 617).
- Additional clarifications
Our manuscript provides a balanced and up-to-date review of glioma modeling technologies. With your suggestions, we are confident the manuscript remains both distinctive and relevant.
Thank you for your valuable insights.
Round 2
Reviewer 2 Report
Comments and Suggestions for Authors
The changes that have been made have greatly improved the quality of this review manuscript.